

# IL-17A exacerbates psoriasis in a STAT3 overexpressing mouse model

Xinran Xie[1,2], Lei Zhang[1,2], Yan Lin[1,2], Xin Liu[1,2], Ning Wang[1,2] and Ping Li[1,2]

[1] Beijing Hospital of Traditional Chinese Medicine, Capital Medical University, Beijing, China
[2] Beijing Institute of Chinese Medicine, Beijing, China

## ABSTRACT

**Background:** Psoriasis is an autoimmune skin disease characterized by immunocyte activation, excessive proliferation, and abnormal differentiation of keratinocytes. Signal transducers and activators of transcription 3 (STAT3) play a crucial role in linking activated keratinocytes and immunocytes during psoriasis development. T helper (Th) 17 cells and secreted interleukin (IL)-17A contribute to its pathogenesis. IL-17A treated STAT3 overexpressing mouse model might serve as an animal model for psoriasis.

**Methods:** In this study, we established a mouse model of psoriasiform dermatitis by intradermal IL-17A injection in STAT3 overexpressing mice. Transcriptome analyses were performed on the skin of wild type (WT), STAT3, and IL-17A treated STAT3 mice. Bioinformatics-based functional enrichment analysis was conducted to predict biological pathways. Meanwhile, the morphological and pathological features of skin lesions were observed, and the DEGs were verified by qPCR.

**Results:** IL-17A treated STAT3 mice skin lesions displayed the pathological features of hyperkeratosis and parakeratosis. The DEGs between IL-17A treated STAT3 mice and WT mice were highly consistent with those observed in psoriasis patients, including S100A8, S100A9, Sprr2, and LCE. Gene ontology (GO) analysis of the core DEGs revealed a robust immune response, chemotaxis, and cornified envelope, et al. The major KEGG enrichment pathways included IL-17 and Toll-like receptor signaling pathways.

**Conclusion:** IL-17A exacerbates psoriasis dermatitis in a STAT3 overexpressing mouse.

## INTRODUCTION

Psoriasis is a genetically determined chronic inflammatory skin disease mediated by the innate and adaptive immune system. It is characterized by erythema, scales, punctate bleeding, and inflammatory infiltration. The pathogenesis of psoriasis is complicated and yet to be deciphered. This condition arises from the activation of immune-mediated T helper (Th) 1/Th17/Th22 cells, which leads to excessive hyperplasia and abnormal differentiation of keratinocytes releasing inflammatory cytokines (*Egeberg et al., 2020*).

Signal transducers and activators of transcription (STAT) three have emerged as an essential factor in various biological processes, including cell proliferation, survival, and

Corresponding author
Ping Li, liping@bjzhongyi.com

migration. *Sano et al. (2004)* demonstrated that STAT3 links activated keratinocytes and immunocytes in the development of psoriasis. They constructed a K5. STAT3C transgenic mouse that exhibits skin lesions resembling human psoriasis. This animal model was used to topically administer 12-O-tetra-decanoylphorbol-13-acetate (TPA) to the dorsal skin resulting in psoriasiform skin lesions that closely resemble psoriatic patients. These findings are crucial factors in the development of psoriasis (*Vinita et al., 2012*). STAT3 is not only a critical factor for T lymphocyte differentiation, but serves as an essential factor in keratinocyte hyperplasia (*Calautti, Avalle & Poli, 2018*). Additionally, their study demonstrated that activated T cells were injected intradermally into K5.STAT3C mice resulted in psoriatic lesions and more aberrant parakeratosis. K5.STAT3C mice and other psoriasiform transgenic mouse models, such as K5.Tie2, K14. vascular endothelial growth factor (VEGF), K5. transforming growth factor (TGF)-β1 transgenic mouse and imiquimod induced mouse models were compared by whole-genome transcriptional profiling to human psoriatic skin. The profiling study supported the value of the K5. STAT3C mouse model as a research tool for investigating psoriasis due to its convergence with human psoriasis (*Swindell et al., 2011*).

Increasing evidence suggests that the interleukin (IL)-23/IL-17 axis plays a central pathogenic role in the development of psoriasis (*Sakkas & Bogdanos, 2017*). IL-17 is the predominant cytokine produced by Th17 cells. The IL-17 family members include IL-17A, IL-17B, IL-17C, IL-17D, IL-17E, and IL-17F. Among them, IL-17A and IL-17F are isoforms that form homodimers or heterodimers with each other. STAT3 genes regulate the differentiation of naive CD4+T cells to Th17 cells (*Tripathi et al., 2017*). Th17 cells release various cytokines, especially IL-17A, which activates intracellular signaling pathways in keratinocytes such as the STAT3 pathway responsible for epidermal hyperplasia (*Shi et al., 2011*). The study crossing the IL-17A[ind] allele to the K14-Cre allele showed that K14-IL-17A[ind/+] mice displayed psoriasis-like features, including hyperkeratosis, parakeratosis, especially exacerbated neutrophil microabscess formation (*Croxford et al., 2014*).

Here, we established a psoriasis mouse model by intradermal injection of recombinant IL-17A into the dorsal skin of STAT3 transgenic mice. Transcriptome analysis was performed to determine the pathogenic characteristics of IL-17A treated STAT3 overexpressing mice.

## MATERIALS AND METHODS

### Animals and experimental design

We established STAT3 overexpressing transgenic mouse with reference to literature (*Sano et al., 2004*), which differs in that STAT3 is not overexpressed in keratinocytes but in systemic tissues. STAT3 plasmid was synthesized from Invitrogen (Invitrogen, Life Technologies Corp., Carlsbad, CA, USA). STAT3 transgenic mouse were made by the Institute of Laboratory Animal Sciences, CAMS&PUMC. The STAT3 cDNA was transferred into the male prokaryotic cells and utilized for microinjection-mediated fertilization of ova. The fertilized cells were transplanted into pseudo-pregnant female mice. The STAT3 gene was identified using polymerase chain reaction (PCR). STAT3

primer sequences were as follows: STAT3-L: 5′-GAGAGTCAAGACTGGGCATATGC-3′, STAT3-R: 5′-CCAGCTCACTCACAATGCTTCTC-3′, 550 bp (NM_0114686.4). The PCR reaction conditions were: 94 °C 5 min, (95 °C 30 s, 55 °C 30 s) × 30 cycles, and 72 °C for 5 min. STAT3 protein expression in the dorsal skin of transgenic mice were verified by western blot (Fig. S1).

Positive STAT3 transgenic mice aged 16–18 weeks of age were used in all experiments. Mice were fed standard laboratory chow and supplied drinking water *ad libitum*. All animal experiments were performed in accordance with the Guidelines for the Care and Use of Laboratory Animals published by the US National Institutes of Health (NIH), all experimental procedures were approved by the appropriate animal welfare committee of the Beijing Hospital of Traditional Chinese Medicine affiliated to Capital Medical University (No. 2017120101).

The experiment was conducted with three groups: wild type (WT) (C57BL/6J strain) group, STAT3 mice group, and IL-17A treated STAT3 mice group. All the dorsal skin of the mice was shaved. The following day, Mice in IL-17A treated STAT3 group were given an intradermal injection of recombinant IL-17A (Peprotech, San Diego, CA, USA) at a dose of 100 µg/kg. At the same time, equal volumes of PBS were intradermally injected into WT mice and STAT3 mice ($n = 10$ per group). After 24 h, all mice were euthanized, and skin samples were harvested and stored at −80 °C for further processing.

## RNA-seq and data analysis

Total RNA was extracted from skin samples using the Qiagen RNeasy columns according to the manufacturer's protocol. RNA concentrations and purity were measured using the NanoDrop 2000 Spectrophotometer (Thermo Fisher Scientific, Wilmington, DE, USA). RNA integrity was assessed using the RNA Nano 6000 Assay Kit run on the Agilent Bioanalyzer 2100 System (Agilent, CA, USA). RNA samples of good quality were used for downstream processing and analysis. All samples were cDNA synthesized, and size was measured using the Agilent 2100. The library's concentration was performed through qPCR, while the index-coded samples were clustered using TruSeq PE Cluster Kitv3-cBot-HS on a cBot Cluster Generation System (Illumina, Santiago, CA, USA). The library was prepared following the sequencing of cluster generation on an Illumina Hiseq2500 (Illumina, Santiago, CA, USA) platform and paired-end reads were generated. The fragments per kilobase of exon per million fragments mapped (FPKM) value represents gene expression. Heatmap was plotted by an online platform for data analysis and visualization (http://www.bioinformatics.com.cn/).

## Differential expressed genes (DEGs) analysis

The DESeq R package (version 1.10.1) was utilized for differential gene expression analysis, with the Benjamini-Hochberg method employed to adjust *P*-values and control false positive rates. Genes exhibiting an absolute log2 (Fold change) greater than or equal to 1.2 as determined by DESeq and adjusted *P*-value less than 0.01 were designated as differentially expressed genes (DEGs). The Venn diagram depicting the overlap of DEGs was generated using Venny 2.1. (http://bioinfogp.cnb.csic.es/tools/venny/).

## Bioinformatics analysis

To identify potential core genes, we imported the DEGs ($|\log2FC| \geq 1.2$) from IL-17A-treated STAT3 mice compared to WT mice into the STRING database (https://string-db.org/) for network construction. Core genes were obtained by CytoNCA analysis using cytoscape software 3.9.0 (https://cytoscape.org/) (degree greater than two times the median) (*Tang et al., 2015*). Gene Ontology (GO) biological process (BP) and Kyoto Encyclopedia of Genes and Genomes pathway (KEGG) analysis were performed using DAVID functional annotation web resource (https://david-d.ncifcrf.gov). Pathways were performed with bioinformatics Toolbox (http://www.bioinformatics.com.cn/).

## Histological staining and immunofluorescent assays

The dorsal skin of mice was fixed in 10% buffered formalin, followed by embedding in paraffin for Hematoxylin and eosin (HE) and immunofluorescence staining. Histopathological changes were imaged using the Aperio CS2 scanner (Leica, San Diego, CA, USA). Epidermal thickness was calculated using the Image-Pro Plus software (version 6.0) (Media Cybernetics, Rockville, MD, USA). Paraffin sections were incubated with proliferating cell nuclear antigen (PCNA) (1:800, #13110; CST, Danvers, MA) and involucrin (1:100, ab53112; Abcam, Cambridge, UK) and then subsequently incubated with their corresponding secondary antibodies. The sections were mounted with DAPI Fluoromount-G$^{TM}$ and observed using the Zeiss LSM 710 confocal microscope (Zeiss, Jena, Germany). Images were acquired using excitation wavelengths of 488 and 405 nm, and then merged.

## Quantitative real-time polymerase chain reaction

Skin samples were subjected to RNA extraction by RNeasy Mini Kit, followed by cDNA synthesis. qPCR was performed three or four times on 7500 Real-Time PCR System (Applied Biosystems, Thermo Fisher Scientific, Wilmington, DE, USA) using the QuantiTect SYBR Green RT-PCR Kit (Qiagen, Hilden, Germany). The reaction was carried out under the following conditions: 95 °C for 30 s, followed by 40 cycles of denaturation at 95 °C for 5 s and annealing/extension at 60 °C for 40 s. To standardize the gene expression levels, we used the $2^{-\Delta\Delta Ct}$ method and normalized them to β-actin. The sequences of qPCR primers are provided in Table S1.

## Statistical analysis

Epidermal thickness and DEGs were presented as "mean ± SEM". Statistical processing was performed using a one-way analysis of variance (one-way ANOVA) by IBM SPSS 26.0 Software. Homogeneity of variance was evaluated by Least-Significant Difference (LSD), and Heterogeneity of variance was evaluated by non-parametric ANOVA (Kruskal–Wallis) was used to compare three groups. Differences were considered statistically significant at $^*P < 0.05$.

## RESULTS

### Evaluating skin phenotypes in WT, STAT3 and IL-17A treated STAT3 mice

There was no significant difference in dorsal skin morphology of the dorsal skin between STAT3 and WT mice. However, IL-17A treated STAT3 mice dorsal skin showed erythema and flaky scaling (Fig. 1A). By histopathology, the skin of IL-17A treated STAT3 mice showed significantly epidermal thickening and dermal inflammatory infiltration compared to STAT3 mice. PCNA, which means the cell is in the mitotic phase, its number in the epidermis's basal layer increased. The involucrin expression up-regulated in the stratum corneum represented parakeratosis (Figs. 1B and 1C). STAT3 and IL-17A treated STAT3 mice significantly differed in epidermal thickening compared to WT mice (Fig. 1D).

### Alterations in DEGs in WT, STAT3 and IL-17A treated STAT3 mice

Cluster analysis of the differentially regulated transcripts was shown in the heatmap image (Fig. 2A). The Illumina HiSeq sequence reads generated in this study have been deposited at BioProject accession number PRJNA431348 in the NCBI-SRA database. The Venn diagram depicted overlaps of significant DEGs among the three comparisons indicated (Fig. 2B).

### The major up-regulated DEGs in pairwise comparison and validation

The top 10 up-regulated DEGs in each pairwise comparison's top 10 are presented in Table 1. The DEGs between WT and IL-17A treated STAT3 mice included S100a protein family such as S100a8 (calgranulin A, MRP-8) and S100a9 (calgranulin B, MRP-14), small proline-rich protein 2 (Sprr2) family such as Sprr2e, Sprr2g and Sprr2d, late cornified envelope-3 (LCE) genes such as LCE3d, LCE3e and LCE3f, et al. Some of the DEGs found in STAT3 mice compared with WT mice have been previously reported to be involved in psoriasis, such as IL-1β and chemokine (C-C motif) ligand (CCL) 4.

DEGs (S100a protein, Sprr2 and LCE genes) from pairwise comparisons of skin samples from WT, STAT3 and IL-17A treated STAT3 mice were validated using qPCR. We selected eight genes for validation, which included S100a8, S100a9, Sprr2e, Sprr2g, Sprr2d, LCE3d, LCE3e, LCE3f. The validation results were concordant with RNAseq data for the DEGs. The expression of S100a8, S100a9, Sprr2e, Sprr2d, LCE3d, and LCE3e in IL-17A treated STAT3 was significantly up-regulated compared to WT mice. Meanwhile, the gene expressions of S100a8, Sprr2d, and LCE3d in IL-17A treated STAT3 mice increased than those of STAT3 mice (Fig. 3).

### Functional enrichment analysis

Protein-protein interaction network construct from the DEGs ($|log2FC| \geq 1.2$) in IL-17A treated STAT3 mice compared to WT mice by the STRING database. The nodes represent proteins. Edges stand for protein-protein associations. Forty-nine core genes were analyzed by CytoNCA in cytoscape software 3.9.0 (Figs. 4A and 4B). Then core DEGs functional analysis and Gene Ontology (GO) analysis were performed by DAVID database. GO terms for biological processes (BP), cellular component (CC), and molecular

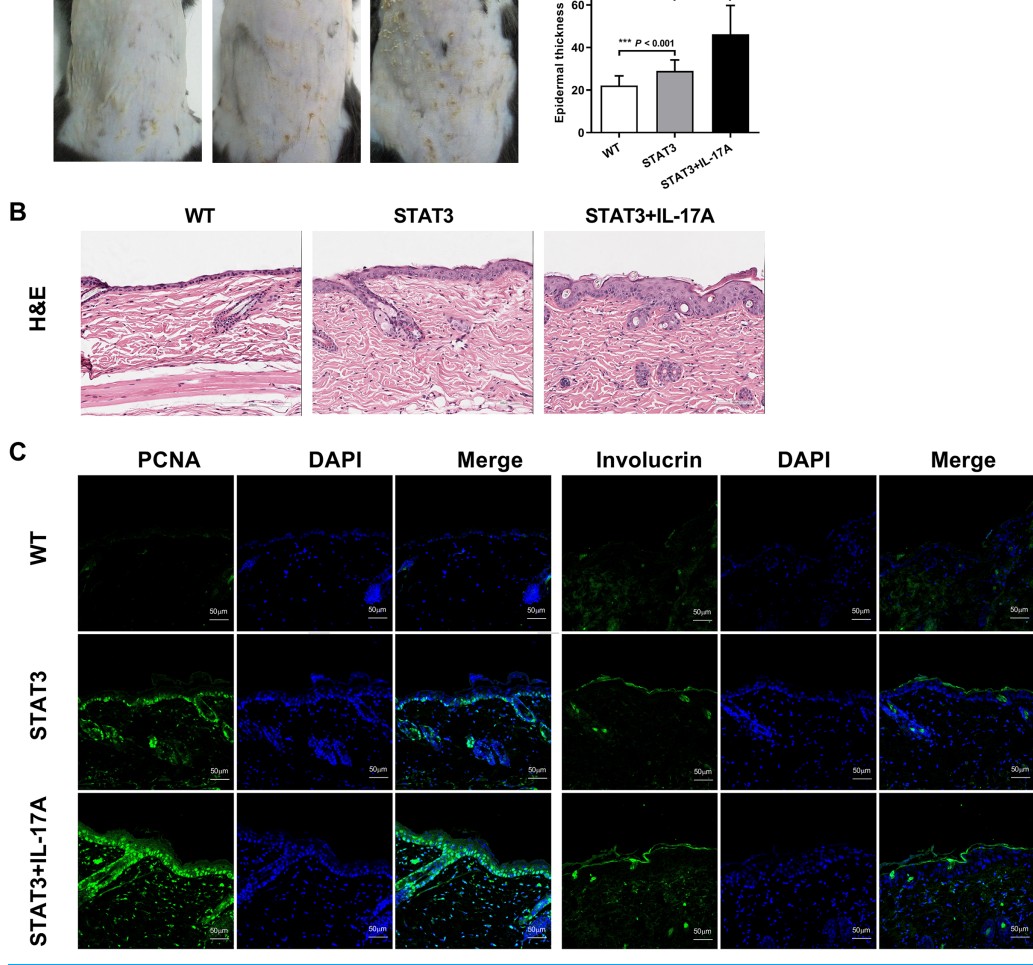

**Figure 1 Skin phenotypes in WT, STAT3, and IL-17A treated STAT3 mice.** STAT3 mice and IL-17A treated STAT3 mice *vs.* wild type (C57BL/6) mice, respectively (*n* = 10). (A) Images of psoriasiform skin. (B) Skin biopsies were stained with hematoxylin and eosin (H&E). Mitotic basal cells are shown in the insets. Scale bar = 200 μm. Representative PCNA and involucrin immunostained images of the dorsal skin. Scale bar = 50 μm. (C) Epidermal thickness was measured using the image analysis system. Differences were considered statistically significant at *$P < 0.05$, **$P < 0.01$, and ***$P < 0.001$.

function (MF) were utilized. Among the BP, DEGs were mainly concentrated in 'immune system process' and 'neutrophil chemotaxis'. CC showed significant enriched 'extracellular region', 'cornified envelope' and 'keratin filament'. MF such as 'cytokine/chemokine activity' exhibited notable enrichment (Fig. 4C).

The enriched pathways identified through cluster analysis of the differentially expressed genes (DEGs) are presented in Fig. 4D. Notably, the most significant pathways include 'IL-17 signaling pathway', 'Coronavirus disease-COVID-19', 'Viral protein interaction with cytokine and cytokine receptor', 'Lipid and atherosclerosis', 'Toll-like receptor signaling pathway', 'C-type lectin receptor signaling pathway', 'NOD-like receptor signaling pathway', 'Cytokine-cytokine receptor interaction', 'NF-kappa B signaling pathway',

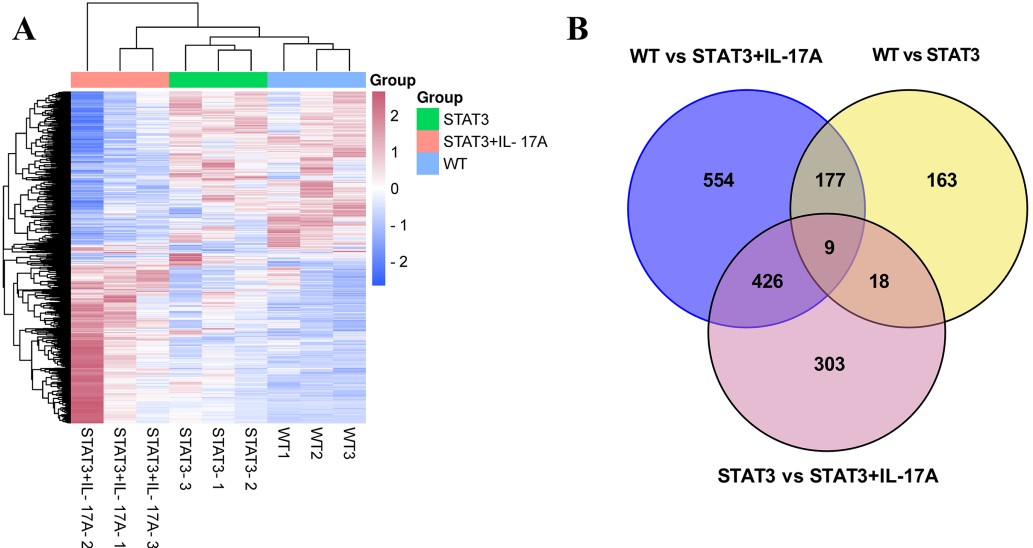

**Figure 2 Differential gene expression in WT, STAT3, and IL-17A treated STAT3 mice.** (A) Hierarchical clustering of gene expressions from the skin of WT, STAT3, and IL-17A treated STAT3 mice was drawn based on the Euclidean distances (FDR < 0.05, log2FC > 1.5-fold difference) ($n$ = 3). (B) Venn diagram depicts the overlap of DEGs between each pairwise comparison.

**Table 1 The top 10 up-regulated DEGs in each pairwise comparison.**

|  | WT *vs.* STAT3 | | WT *vs.* STAT3+IL-17A | | STAT3 *vs.* STAT3+IL-17A | |
|---|---|---|---|---|---|---|
|  | Gene symbol | logFC | Gene symbol | logFC | Gene symbol | logFC |
| 1 | Mup1 | 2.71 | Sprr2e | 7.56 | Gm5414 | 4.38 |
| 2 | Clec4e | 2.68 | Sprr2g | 6.32 | Sh2d5 | 4.18 |
| 3 | Mup7 | 2.53 | Chil3 | 6.05 | Lcn2 | 3.13 |
| 4 | Il1b | 2.46 | S100A8 | 6.04 | Krt84 | 3.11 |
| 5 | Irg1 | 2.27 | Gm5414 | 5.36 | Ca4 | 2.99 |
| 6 | Ccl4 | 2.25 | Lce3d | 5.34 | Sprr2b | 2.96 |
| 7 | Mup14 | 2.16 | S100A9 | 5.28 | Saa3 | 2.90 |
| 8 | Trem3 | 2.15 | Lce3f | 5.12 | Sprr1b | 2.86 |
| 9 | Mup19 | 2.15 | Lce3e | 5.00 | Sprr2h | 2.82 |
| 10 | Trim30b | 2.15 | Sprr2d | 4.94 | Timp1 | 2.74 |

'Chemokine signaling pathway' and 'TNF signaling pathway'. The main enriched pathways, namely the IL-17 signaling pathway and Toll-like receptor signaling pathway, demonstrate the primary pathogenesis mechanism of psoriasis by involving significant genes in these pathway (Figs. 4E and 4F).

## Confirmation of core DEGs by qPCR

The core DEGs identified from CytoNCA and functional enrichment analysis were confirmed using qPCR. Eight genes, including TNF-α, IL-1β, chemokine (C-X-C motif)

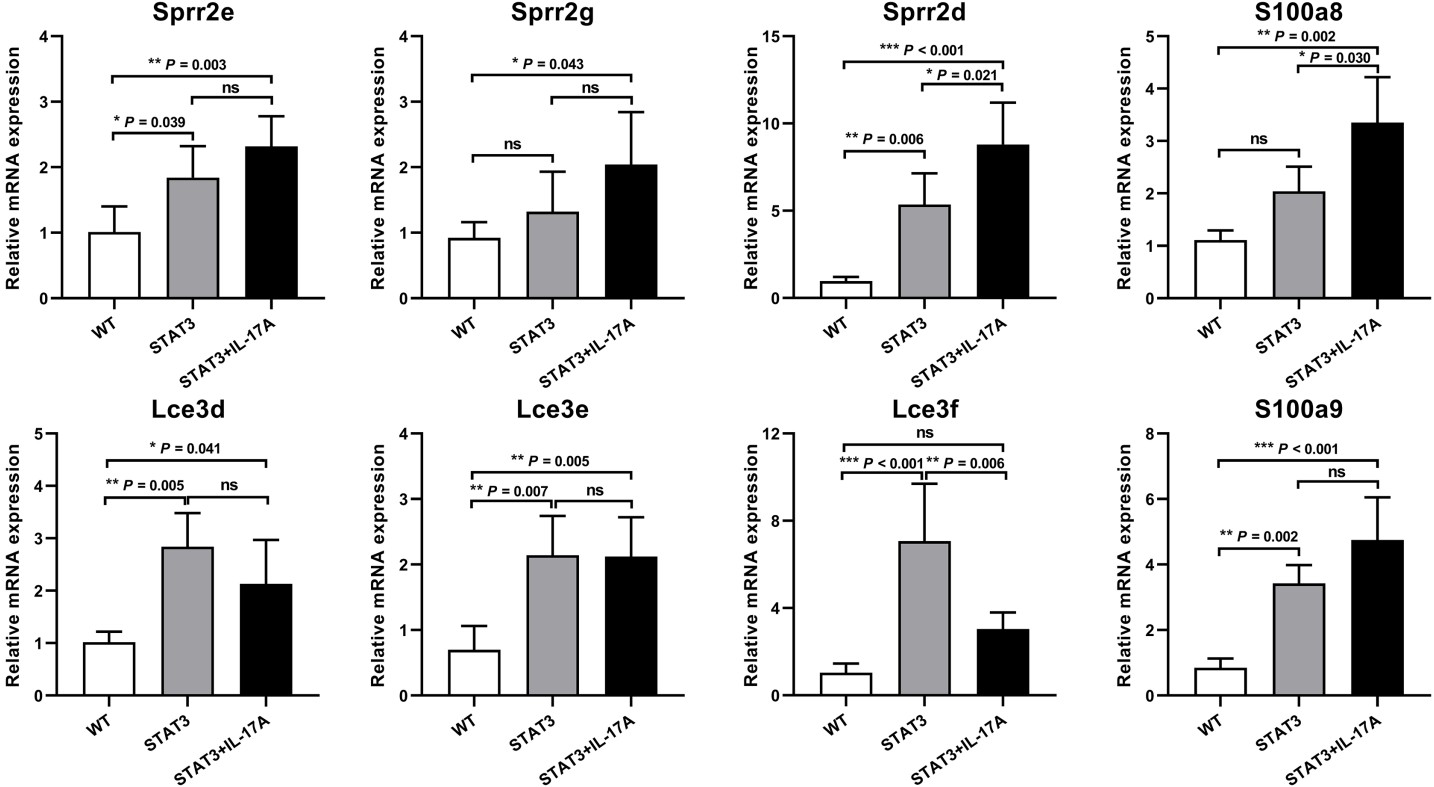

**Figure 3 qPCR validation of the top ten up-regulated genes in WT, STAT3, and IL-17A treated STAT3 mice ($n = 3$).** Eight mRNAs were selected for qPCR validation of the sequencing analysis data. The expressions of S100A8, S100A9, Sprr2e, Sprr2d, LCE3e, and LCE3d in IL-17A treated STAT3 mice were significantly up-regulated compared to WT mice. The expressions of S100a8, Sprr2d, and LCE3d in IL-17A treated STAT3 mice increased than those of STAT3 mice. The ratio of each mRNA relative to β-actin was calculated using the $2^{-\Delta\Delta Ct}$ method. Differences were considered statistically significant at $^*P < 0.05$, $^{**}P < 0.01$, and $^{***}P < 0.001$.

ligand (CXCL) 1, CXCL2, CCL3, CCL4, CD14, and TLR7 were selected for validation. The gene expressions of TNF-α, IL-1β, CXCL1, CXCL2, CCL4, and TLR7 in STAT3 mice treated by IL-17A were significantly up-regulated compared to WT mice. Additionally, the expressions of TNF-α, IL-1β, CXCL1, CXCL2, CCL3, CCL4, and TLR7 in IL-17A treated STAT3 mice increased than those of STAT3 mice (Fig. 5).

## DISCUSSION

Psoriasis is a chronic inflammatory disease characterized by immune-mediated excessive proliferation and differentiation of keratinocytes. In this study, we referred to Sano's K5. stat3C transgenic animal model, which differs in that STAT3 is not overexpressed in keratinocytes but in systemic tissues. Subsequently, we established an IL-17A treated STAT3 overexpressing mouse model, exhibiting typical pathological psoriasis features such as abnormal proliferation and differentiation of epidermal cells. The increased expression of PCNA and involucrin suggests the presence of hyperkeratosis and parakeratosis, the typical pathological features commonly observed in psoriasis. Among the DEGs identified between IL-17A treated STAT3 mice and WT mice were S100a8, S100a9, Sprr2 protein, and Lce genes, all previously associated with psoriatic patients

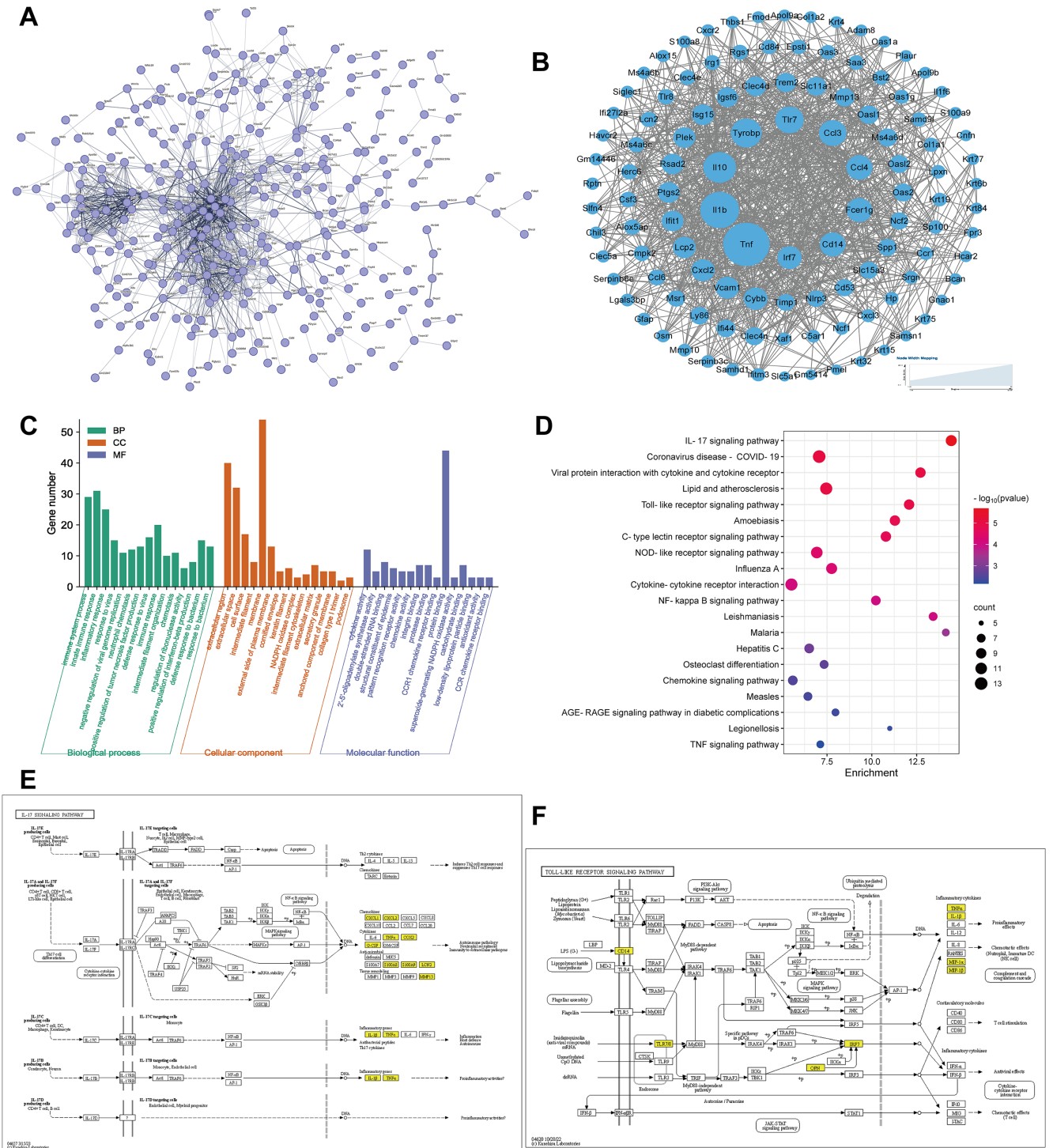

**Figure 4 Bioinformatics analysis of DEGs between IL-17A treated STAT3 mice and WT mice.** (A) Protein-protein interaction networks analysis. The network construct from the DEGs (|log2FC| ≥ 1.2) in IL-17A treated STAT3 mice compared to WT mice. The nodes represent proteins. Edges stand for protein-protein associations (hide disconnected nodes in the network). (B) Forty-nine core genes were obtained by CytoNCA analysis (degree greater than two times the median) using Cytoscape (www.cytoscape.org). (C–D) GO analysis and KEGG pathway enrichment analysis of DEGs using the online DAVID database (https://david-d.ncifcrf.gov). (C) GO enrichment analysis for biological process, cellular component, and molecular function. (D) KEGG pathway enrichment analysis. (E) IL-17 signaling pathway. (F) Toll-like receptor signaling pathway. Yellow boxes are represented upregulated DEGs detected by RNA-seq.

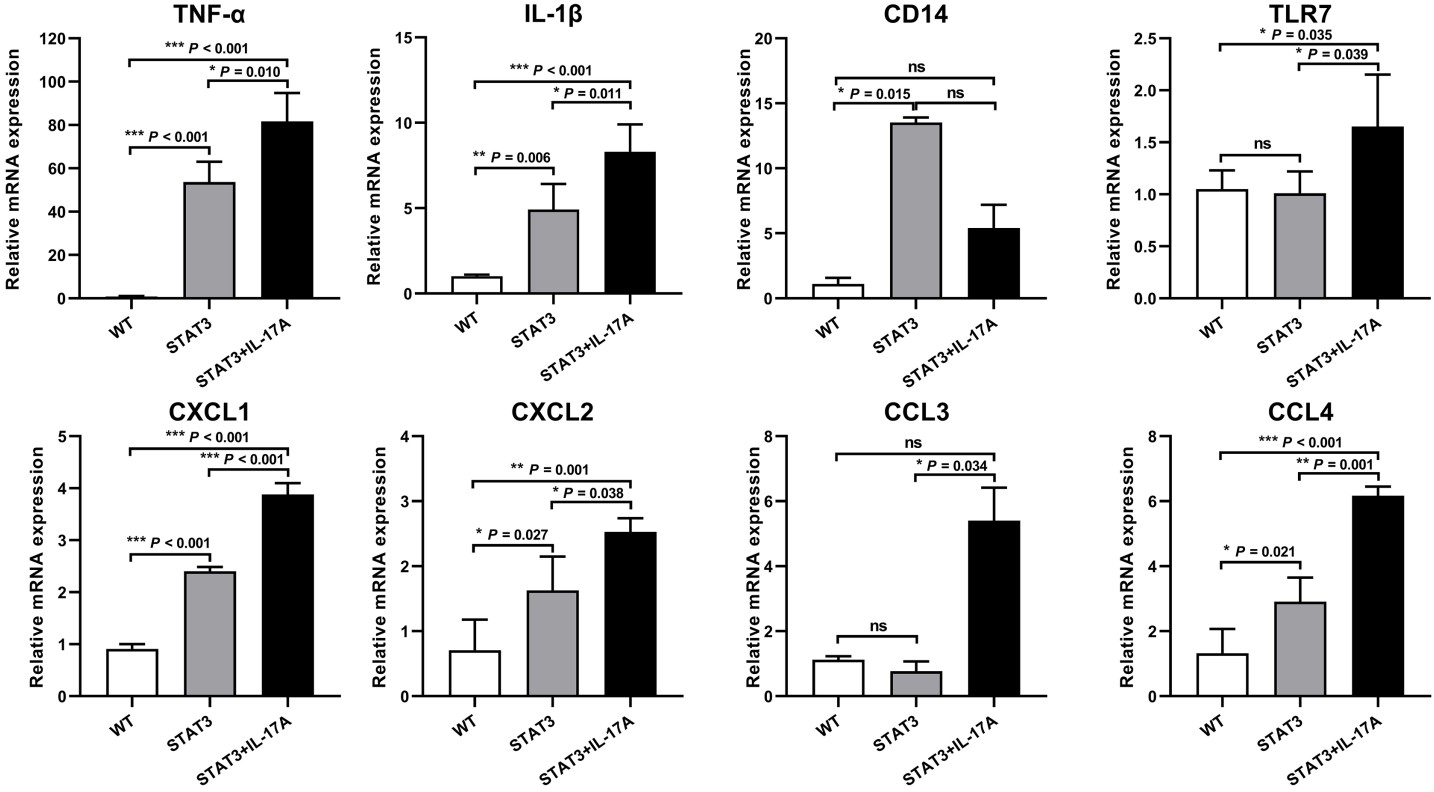

**Figure 5** **qPCR validation of the core genes in WT, STAT3, and IL-17A treated STAT3 mice (*n* = 3).** Eight mRNAs were selected for qPCR validation of the enrichment pathway. The expressions of TNF-α, IL-1β, CXCL1, CXCL2, CCL4, and TLR7 in IL-17A treated STAT3 mice were significantly up-regulated compared to WT mice. The expressions of TNF-α, IL-1β, CXCL1, CXCL2, CCL3, CCL4, and TLR7 in IL-17A treated STAT3 mice increased than those of STAT3 mice. The ratio of each mRNA relative to β-actin was calculated using the $2^{-\Delta\Delta Ct}$ method. Differences were considered statistically significant at $^{*}P < 0.05$, $^{**}P < 0.01$, and $^{***}P < 0.001$.  

(*Bhattacharya et al., 2018*; *D'Erme et al., 2015*). The expressions of core genes, including TNF-α, IL-1β, CXCL1, CXCL2, CCL3, CCL4, CD14, and TLR7 were upregulated in IL-17A treated STAT3 mice. Bioinformatics analysis of the core DEGs enriched for the IL-17 signaling pathway, immune inflammation, cell chemotaxis, and keratinization that reflected the pathological features of psoriasis.

Psoriatic hallmark features are characterized by epidermal acanthosis, hyperkeratosis, and parakeratosis. It has been established that IL-17A is involved in keratinocyte differentiation (*Pfaff et al., 2017*). Our study revealed a dozen epidermal differentiation complex (EDC) genes that were differentially expressed following injection of IL-17A into STAT3 mice. The EDC contains several genes that include loricrin, involucrin, S100a calcium-binding proteins, Sprr2 proteins and Lce genes (*Mischke et al., 1996*). They are located on human chromosome 1q21 with character traits that are involved in epidermal differentiation (*Motomu et al., 1997*). These gene clusters are located on mouse chromosome 3, exhibit similar differentiation functions to those observed in humans. S100a8 and S100a9 are known for their antimicrobial function and are expressed in differentiated keratinocytes, where they co-localize with differentiation markers such as involucrin (*Schmidt et al., 2001*). In addition, S100a8 and S100a9 are potent

pro-inflammatory mediators and are triggered by IL-17A (*Jin et al., 2014*; *Podgorska et al., 2018*). Sprr2 encodes for a family of cornified cell envelope precursor proteins and is strongly induced during psoriatic hyperproliferation (*Hohl et al., 1995*). The Sprr2 genes family in mice consists of 11 isoforms (Sprr2a–2k) that exhibit differential expression patterns in epithelial tissues. Sprr2 was found to be non-coordinately up-regulated by IL-6/gp130/STAT3 signaling and functions in cell migration and wound healing (*Lambert et al., 2017*; *Nozaki et al., 2004*). The Lce gene cluster is a psoriasis susceptibility identified through genome-wide analysis (*Zhang et al., 2009*). Lce genes encode stratum corneum proteins of the cornified envelope that function in epidermal terminal differentiation and are modulated by Th17 cytokines (*Bergboer et al., 2011*). We found Lce3d-f transcript to be overexpressed in both STAT3 and IL-17 treated STAT3 mice. Previous studies have reported a strong association between the severity of psoriasis and the Lce3a and Lce3d locus (*Julia et al., 2012*). While compensatory up-regulation of Lce3e has been documented to counteract deletions in LCE3a/b/c/d, the role of Lce3f remains unreported (*Karrys et al., 2018*). Our findings indicate terminal differentiation in granular keratinocytes and incomplete squamous corneocyte formation in IL-17 treated STAT3 mice, resulting in characteristic psoriasis lesions.

The core DEGs identified by CytoNCA analysis were inflammatory cytokines (TNF-$\alpha$ and IL-1$\beta$) and chemokines (CXCL1, CXCL2, CCL3, and CCL4), which were mainly secreted by IL-17A-induced keratinocytes. These DEGs were confirmed to be highly upregulated in IL-17A treated STAT3 mice. TNF-$\alpha$ is a proinflammatory cytokine produced by multiple cells, including Th17 cells and keratinocytes. It amplifies inflammation by inducing the expression of other proinflammatory cytokines and neutrophil chemokines, promoting vascular adhesion molecules to facilitate the influx of inflammatory cells, and amplifying the effects of other cytokines such as IL-17 (*Chiricozzi et al., 2011*). TNF-$\alpha$ inhibitors or monoclonal antibodies have been proven effective in treating psoriasis (*Yiu et al., 2022*). IL-1$\beta$ is a well-known inducer and effector of inflammation. In psoriasis, IL-1$\beta$ is mediated by an apoptosis-associated speck-like protein containing a caspase recruitment domain, such as NLRP3 and AIM2 (*Zwicker et al., 2017*). CXCL1 is a neutrophil-activating protein-3 derived from keratinocytes. CXCL2 is also called macrophage inflammatory protein-2a (MIP-2$\alpha$). CXCL1 and CXCL2 recruitment to sites of inflammation by binding to CXC chemokine receptor 2 (CXCR2) (*Sellau & Groneberg, 2020*). Stimulated leukocytes, fibroblasts, and tumor cells produce CCL3 (MIP-1$\alpha$) and CCL4 (MIP-1$\beta$), which induce chemotaxis of T cells, monocytes, NK cells, and dendritic cells by interacting with their specific receptor CCR5. These chemokines have diverse effects on various immune and non-immune cells (*Yazdani et al., 2020*). CD14 acts as a co-receptor for several other toll-like receptors (TLRs), including TLR2, TLR4, TLR7, and TLR9. The upregulation of CD14 gene expression in pathological epidermis leads to enhanced inflammatory signaling and compromised epidermal barrier function (*Dolivo et al., 2022*).

The gene ontology analysis revealed significant enrichment of immune system process and chemotaxis in the biological process category, extracellular region, cornified envelope, and keratin filament in the cellular component category, as well as cytokine and

chemokine activity in the molecular function category. These findings are consistent with the pathological features of psoriasis (*Mehta et al., 2017*). KEGG pathway analysis identified significant enrichment of the IL-17 signaling pathway and toll-like receptor signaling pathway among DEGs. The enrichment for pathways and terms was similar to RNA sequencing of skin biopsy samples from psoriasis patients and healthy controls (*Dou et al., 2017*; *Niehues et al., 2017*). TLRs are generally upregulated, which promotes the transcription and release of proinflammatory cytokines (*Chen, Szodoray & Zeher, 2016*). Furthermore, DEGs were found to be clustered for lipid and atherosclerosis, suggesting a potential mechanistic link between psoriasis and an increased risk of cardiovascular diseases (*Hu & Lan, 2017*; *Li et al., 2014*).

This study has certain limitations and deficiencies. The absence of IL-17A treated WT mice in our experimental design is a limitation. There were limited studies on establishing a psoriasis model through IL-17A injection, in contrast to IL-23 which has become a classic model for psoriasis due to its ability to stimulate Th17 cell activation *via* secretion from antigen-presenting cells. It was reported that subcutaneous injection of 3 µg recombinant mouse IL-17A in 100 µl PBS into C57BL/6 mice significantly increased CXCL1 in plasma, with no pathological change images presented (*Lipovsky et al., 2021*). In the study, we applied a lower dose of 100 µg/kg (about 2 µg per mouse) on STAT3 overexpressing mice, and focused on differential gene expression in the skin lesions of mice. We observed a significant increase in inflammatory factors (such as TNF-α, IL-1β, CXCL1, CXCL2, CCL4, and TLR7) and exacerbates hyperkeratosis and keratosis in STAT3 mice.

## CONCLUSIONS

Therefore, our study established an animal model by intradermal IL-17A injection in STAT3 transgenic mice and verified that IL-17A exacerbates psoriasis dermatitis in a STAT3 overexpressing mouse.

### Funding

This work was supported by the National Natural Science Foundation of China (No. 81873119, 81603630), and the foundation of Beijing Institute of Chinese Medicine. The funders had no role in study design, data collection and analysis, decision to publish, or preparation of the manuscript.

### Grant Disclosures

The following grant information was disclosed by the authors:
National Natural Science Foundation of China: 81873119 and 81603630.
Foundation of Beijing Institute of Chinese Medicine.

### Competing Interests

The authors declare that they have no competing interests.

## Author Contributions

- Xinran Xie conceived and designed the experiments, performed the experiments, analyzed the data, prepared figures and/or tables, authored or reviewed drafts of the article, and approved the final draft.
- Lei Zhang performed the experiments, analyzed the data, prepared figures and/or tables, and approved the final draft.
- Yan Lin performed the experiments, prepared figures and/or tables, and approved the final draft.
- Xin Liu performed the experiments, prepared figures and/or tables, and approved the final draft.
- Ning Wang performed the experiments, prepared figures and/or tables, and approved the final draft.
- Ping Li conceived and designed the experiments, authored or reviewed drafts of the article, and approved the final draft.

## Animal Ethics

The following information was supplied relating to ethical approvals (*i.e.*, approving body and any reference numbers):

Animal welfare committee of Beijing Hospital of Traditional Chinese Medicine affiliated to Capital Medical University approval for this research (No. 2017120101).

## DNA Deposition

The following information was supplied regarding the deposition of DNA sequences:

The RNA sequences are available at NCBI-SRA: PRJNA431348.

## Data Availability

The raw measurements are available in the Supplemental Files.

## Supplemental Information

Supplemental information for this article can be found online at http://dx.doi.org/10.7717/peerj.15727#supplemental-information.

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
