# Peer review of "IL-17A exacerbates psoriasis in a STAT3 overexpressing mouse model"

_PeerJ, doi:10.7717/peerj.15727_

## Round 0.1 · original submission · Major Revisions

Please systematically review the comments on the reviewers and address these where possible. Where not possible, please let me know, explaining why. It will be essential to clarify the hypothesis of this work and ensure all data images are of sufficient quality/resolution, and how these compare with other models of psoriasis. Please attend to the PeerJ raw data policy.

·

Basic reporting

Clear, unambiguous, professional English language used throughout.
• The language in this manuscript needs to be improved for clarity in a number of areas. Most particularly the description of the transgenic STAT3 overexpressing mice which is inconsistent throughout. In addition, there are numerous phrases which are not grammatically sound, examples of which are highlighted throughout the text. For instance, L19-20 should read “transgenic STAT3 overexpressing mice were treated intradermally with IL-17” and thereafter these mice should be referred to as “IL17 treated STAT3 overexpressing mice” or equivalent
Intro & background to show context.
• The introduction and background are solid and give the context for this work.
Literature well referenced & relevant.
• The literature cited is meaningful and in the correct context
Structure conforms to PeerJ standards, discipline norm, or improved for clarity.
• The structure of the paper is straightforward to follow but there are several pieces of text that require improvement
Figures are relevant, high quality, well labelled & described.
• Several figures contain errors or are not well annotated.
• Fig S1: A - the lanes are no labelled and B - transgenic is misspelled
• Figure 1: Image quality for panels A and B is low, is this a resolution issue in the reviewer platform or are the images just low quality?
• Figure 4: the images are of such low resolution as to make them uninterpretable without the text

Raw data supplied (see PeerJ policy).
• Raw data is not supplied for the transcriptomic analysis in Fig 2

Experimental design

Original primary research within Scope of the journal.
• Yes
Research question well defined, relevant & meaningful. It is stated how the research fills an identified knowledge gap.
• The research question is poorly defined and does not add a lot to the existing knowledge in the field. It is well established that IL17 and STAT3 are linked in psoriasis, and this study adds evidence for one particular model but claims far beyond what the data shows.
• This study is missing a number of critical controls which are crucial for the interpretation of the data.
1. There is no WT+IL17A group to compare the skin in the absence of STAT3 overexpression.
2. There is no data to tell us if STAT3 overexpression is global or limited to specific cell types as in Sano et al. This impacts the interpretation of the data as the implication of their conclusions and Figure 6 schematic is that all IL17 effects via STAT3 are keratinocyte mediated, which we cannot conclude from this data.
3. This paper claims to have uncovered a link between STAT3 and IL17 signalling in psoriasis which is a) not novel (https://doi.org/10.3389/fimmu.2021.621956) and b) not clear from the data. What they do demonstrate is that IL17 may exacerbate psoriasis in this particular STAT3 overexpression model in a manner that is not clear. IL-17 blockade in the STAT3 overexpressing mice would be a much clearer means of demonstrating a causal link between these two mediators in this experimental system.

Rigorous investigation performed to a high technical & ethical standard.
• The evidence presented is not convincing in this regards. The study rests on a STAT3 overexpressing mouse model generated by Sano et al that should express STAT3 in a keratinocyte specific manner. The authors do not demonstrate that this is the case, and the evidence that their mice overexpress STAT3 at all is mediocre. The experiments they perform subsequently add more strength to the case that STAT3 overexpression generates a psoriasis-like phenotype, and that IL-17 exacerbates this via a number of chemical mediators. Figure 2 and Figure 4 use bioinformatic methodologies which are not clearly reported and poorly presented (mostly this is the case for Figure 4).
Methods described with sufficient detail & information to replicate.
• The methods used to generate the STAT3 overexpressing mice are not sufficiently clear. Do these mice overexpress STAT3 in a keratinocyte limited manner as in Sano et al or not? The authors must state this in order for the reader to correctly interpret their results. Figure S1 is very poorly presented as outlined elsewhere in the review.
• Figure 2 and Figure 4 use bioinformatic methodologies which are not clearly reported and poorly presented (mostly this is the case for Figure 4).

Validity of the findings

Impact and novelty not assessed.
• N/A
Meaningful replication encouraged where rationale & benefit to literature is clearly stated.
• There is reference to the paper by Sano et al where this work’s premise originates, but little attempt to replicate its findings as a meaningful beginning to the study. Most importantly, Sano et al describe a K5 linked STAT3 overexpression which limits the nature of the overexpression to keratinocytes. In this study, this is not addressed at all and there is no attempt to show that their STAT3 overexpression is cell type specific at all which has major implications on the interpretation of the data.
All underlying data have been provided; they are robust, statistically sound, & controlled.
• All underlying data have not been provided (see Figure comments above), but what is presented in the paper seems robust. However, the authors are frequently using t-tests rather than 1-way ANOVA to assess experiments with 3 groups which is incorrect and must be addressed.
• This study is missing a number of critical controls which are crucial for the interpretation of the data.
1. There is no WT+IL17A group to compare the skin in the absence of STAT3 overexpression.
2. There is no data to tell us if STAT3 overexpression is global or limited to specific cell types as in Sano et al. This impacts the interpretation of the data as the implication of their conclusions and Figure 6 schematic is that all IL17 effects via STAT3 are keratinocyte mediated, which we cannot conclude from this data.
3. This paper claims to have uncovered a link between STAT3 and IL17 signalling in psoriasis which is a) not novel (https://doi.org/10.3389/fimmu.2021.621956) and b) not clear from the data. What they do demonstrate is that IL17 may exacerbate psoriasis in this particular STAT3 overexpression model in a manner that is not clear. IL-17 blockade in the STAT3 overexpressing mice would be a much clearer means of demonstrating a causal link between these two mediators in this experimental system.
Conclusions are well stated, linked to original research question & limited to supporting results.
• The discussion and conclusion are well referenced and well written, but are not supported by the results presented in the manuscript for the most part. The central claim of the paper is not supported due to the absence of the controls stated above. I believe the paper would be better served to focus more on the effect of IL17 in this particular psoriasis model, and limit its claims to what the data shows (i.e. that IL-17 appears to exacerbate psoriasis in mouse that overexpresses STAT3)

Additional comments

This paper likely contains reasonable data that would be of some value to the field. It should be reworked to increase confidence in the underlying mythology, currently describe the experimental system, and to ensure its claims are more accurately supported by the data.

The title (Transcriptome analysis identiûes activation of IL-17A and STAT3 circuit activation for the pathogenesis of psoriasis) is not supported by the findings. A title along the lines of "IL-17 exacerbates psoriasis in a STAT3 over expressing mouse model" would be much more sensible and, along with methodological improvements, would make the study more credible.

Reviewer 2 ·

Basic reporting

In general, the manuscript is clear and professional English used throughout (pg 8 line 41 – repeated word)
References - References would usually be cited before the full stop at the end of sentences
Article structure is professional and raw data shared and accessible. Figure 4 was not of sufficient quality to view, however the authors kindly provided a clearer version.
Methods: pg 24 line 92 (The next day intradermal were induced with recombinant murine IL-17A…) needs to be rephrased to make sense.
The results sections would benefit from a brief rationale and description of the experiment performed. This would improve the flow and readability of the paper.
Discussion: Pg 10 line 202 (IL-17A induced STAT3 mice dorsal skin) needs to be rephrased

Experimental design

There was no clear research question/hypothesis stated – this needs to be addressed by the authors, ideally in the abstract of the manuscript. Then in the discussion the authors should then link their conclusions to the research question.

Validity of the findings

Figure 1: Could the authors discuss the histological findings from their study in the context of other psoriasis models? Are the psoriatic features as pronounced?

Figure 2A: It is hard to observe any differences between the groups in the heatmap. It appears that what has been shown in the heatmap is perhaps just the normalized counts (this should be clarified). The authors should attempt re-analysing the data with alternate scaling.

Additional comments

In their conclusion the authors state ‘Administration of exogenous IL-17A significantly hyperactivates STAT3 resulting in proliferation and differentiation of keratinocytes. Keratinocytes, in turn, produce some cytokines and chemokines to trigger immune-inflammatory responses and promote Th17 cells to secrete endogenous IL-17A’. In the light of this it seems surprising that IL-17A was not one of the highly upregulated genes in STAT3+IL-17A mice. Can the authors discuss why this might be?

---

## Round 0.2 · accepted · Accept

Thank you for addressing the reviewers' comments to their satisfaction. While it would be been preferable for the controls suggested for Figure 1 be included, we note that this issue has been sufficiently addressed in the Limitations section.

·

Basic reporting

The language has been clarified significantly, the authors must be commended for their efforts here.

Experimental design

I believe the issues highlighted previously have been addressed, either by modifications to the text, statistical analysis or additions to the limitations section of the article.

It is disappointing that the controls suggested for Figure 1 could not be included, but it is understandable and now sufficiently addressed in the Limitations section.

Validity of the findings

The findings are now clarified and better contextualized in the text's discussion, allowing the data and article to be of use to the field. The addition of this data to the psoriasis community may provide the basis for useful studies with this model.

Additional comments

I believe the manuscript is significantly improved and a fairer reflection of the data contained within it. I believe it may be of use to the field in its current format.

Reviewer 2 ·

Basic reporting

No comment

Experimental design

No comment

Validity of the findings

No comments

Additional comments

The authors have addressed my comment adequately